# Postbiotics and Their Health Modulatory Biomolecules

**DOI:** 10.3390/biom12111640

**Published:** 2022-11-04

**Authors:** Emma Scott, Kim De Paepe, Tom Van de Wiele

**Affiliations:** Center for Microbial Ecology and Technology (CMET), Ghent University, 9000 Ghent, Belgium

**Keywords:** postbiotics, gut health, epithelial barrier, microbiome, exopolysaccharides, bacteriocins, SCFAs, butyrate, antioxidant

## Abstract

Postbiotics are a new category of biotics that have the potential to confer health benefits but, unlike probiotics, do not require living cells to induce health effects and thus are not subject to the food safety requirements that apply to live microorganisms. Postbiotics are defined as a “preparation of inanimate microorganisms and/or their components that confers a health benefit on the host”. Postbiotic components include short-chain fatty acids, exopolysaccharides, vitamins, teichoic acids, bacteriocins, enzymes and peptides in a non-purified inactivated cell preparation. While research into postbiotics is in its infancy, there is increasing evidence that postbiotics have the potential to modulate human health. Specifically, a number of postbiotics have been shown to improve gut health by strengthening the gut barrier, reducing inflammation and promoting antimicrobial activity against gut pathogens. Additionally, research is being conducted into the potential application of postbiotics to other areas of the body, including the skin, vagina and oral cavity. The purpose of this review is to set out the current research on postbiotics, demonstrate how postbiotics are currently used in commercial products and identify a number of knowledge gaps where further research is needed to identify the potential for future applications of postbiotics.

## 1. Introduction

It is well established that the human microbiome plays a key role in human health and disease. The human body is host to a number of different microbiomes, including those of the gut, skin, vagina and oral cavity, of which the most researched is the gut microbiome. Until now, the research focus has been to identify the microbes that colonise the different host niches with a view to developing biotic therapies that seek to alter the composition of the microbial communities in order to impact health and disease states, principally antibiotic and probiotic therapies.

Antibiotics have been utilised to alter the composition of various microbiomes, specifically where pathogens are considered to play a key role in the aetiology of a particular disease or disorder. There are two main limitations with antibiotics. Firstly, antibiotics alter the microbiome by destroying microbes using a non-targeted approach, thus destroying beneficial microbes alongside the specific pathogens involved in the disease or disorder. Secondly, after a course of antibiotics, the microbial composition is not always fully restored and, as such, antibiotics can create an unintended persistent microbiome disturbance termed dysbiosis that can lead to further diseases or disorders.

To avoid harmful off-target effects towards the symbiotic microbiome and curb antibiotic resistance, considerable research effort has been made to identify microbes that are associated with a healthy microbiome and that could be used as a probiotic to alter the composition of various microbiomes. Probiotics are “live microorganisms that, when administered in adequate amounts, confer a health benefit on the host” [1]. Current probiotic therapies are impacted by two main limitations that directly impact their effectiveness. Firstly, probiotic therapies are limited to the use of food-safe microbes, namely those microbes that are Generally Recognised as Safe (GRAS) by the US Food and Drug Administration (FDA) or have the Qualified Presumption of Safety (QPS) by the European Food Safety Authority (EFSA) [2]. However, as most endogenous microbes within a healthy human gut microbiome are not considered food-safe, such endogenous microbes are currently not permitted to be used in human application even where there is research evidence that demonstrates they could have health-promoting properties. Secondly, when administering probiotics orally to target the gut microbiome, the main challenge is retaining viability during upper gastrointestinal tract passage as exposure to stomach acids and bile salts may interfere with the targeted delivery of an adequate number of viable cells. As a result of these limitations, orally administered probiotic therapies are not typically successful in colonising the gut microbiome in the long term and thus producing the health benefits suggested in research studies [3].

However, there is increasing evidence that the health benefits associated with probiotics are typically derived from metabolites and secreted compounds produced by the live microbes rather than solely the presence of the live microbes themselves [4]. Furthermore, evidence has also demonstrated health benefits that are associated with the immunomodulatory activity of a number of specific cell constituents (e.g., Microbe-Associated Molecular Patterns or MAMPs) that are part of the cell membrane and/or cell wall and are released upon cell death [5]. Consequently, microbiome modulation research is increasingly focusing on postbiotics [4].

## 2. Evolving Research from Probiotics to Postbiotics

Postbiotics are defined as a “preparation of inanimate microorganisms and/or their components that confers a health benefit on the host” [6]. This is a consensus definition proposed by the International Scientific Association of Probiotics and Prebiotics (ISAPP) in 2021 to replace several previously proposed but inconsistent definitions. Examples are set out in the 2021 ISAPP consensus statement [6] and include: (i) “soluble factors (products or metabolic byproducts), secreted by live bacteria, or released after bacterial lysis, such as enzymes, peptides, teichoic acids, peptidoglycan-derived muropeptides, polysaccharides, cell-surface proteins and organic acids”; (ii) “non-viable metabolites produced by microorganisms that exert biological effects on the hosts”; (iii) “compounds produced by microorganisms, released from food components or microbial constituents, including non-viable cells that, when administered in adequate amounts, promote health and wellbeing” [6].

These definitions were inconsistent with respect to the inclusion or exclusion of non-viable cells or cell components besides the metabolites and/or secreted products of live bacteria. To resolve this disparity, ISAPP established that postbiotics must include an inactivated cellular biomass, comprising the components of an inanimate microbe that remains after a live microbe has been subjected to an inactivation process [6]. In addition, the ISAPP consensus statement clarifies that postbiotic preparations must originate from microbes that are characterised, thus excluding preparations that originate from microbes that are undefined [6]. Aside from this, postbiotic formulations may also contain metabolic and secreted products produced by microbes during growth and fermentation in a medium prior to the inactivation process. These products include short-chain fatty acids (SCFAs), exopolysaccharides (EPS), vitamins, teichoic acids, bacteriocins, enzymes and peptides [5]. Although these non-cellular metabolites and secretions are produced by live microbes during fermentation in a medium or host environment, they are not explicitly included in the definition of probiotics. Consequently, the term “postbiotics” was initially used as a synonym to “cell-free supernatant” or “cell-free filtrate” to refer to the products remaining in a fermentation medium after the removal of cells through centrifugation and filtration, respectively [6]. Similarly, the term “postbiotic component” has been used to refer to an isolated product within a cell-free medium. This sprawl in terms and definitions arising from the lack of a consensus definition of postbiotics is important to consider when searching the literature on this topic prior to 2021.

To date, as most postbiotic research has been an extension of earlier probiotic research efforts, the focus has been on food-safe microbes, principally lactic acid bacteria. However, the stringent regulations restricting probiotic therapies for human application to food-safe GRAS or QPS do not apply to postbiotics, which do not pose a risk of infection, since inactivation precludes the replication of administered postbiotics inside the body [7]. This postbiotic property may open up possibilities to generate postbiotic preparations originating from a considerably more diverse range of microbes, including microbial taxa that lack a history of safe use and are therefore excluded from consideration for probiotic therapies. 

For example *Faecalibacterium prausnitzii*, which is generally recognised as a beneficial endogenous gut microbe, cannot be administered as a live microbe in probiotic therapy as it is currently not deemed safe to consume [8]. However, it may be possible to achieve the same health benefits with postbiotic therapy based on an inactivated *Faecalibacterium prausnitzii* preparation containing butyrate, as studies suggest that the butyrate production by *Faecalibacterium prausnitzii* provides the health promoting function within the gut [8]. In addition to butyrate, *Faecalibacterium prausnitzii* also produces a Microbial Anti-inflammatory Molecule (MAM), a protein which has been demonstrated to have anti-inflammatory effects on mice models with induced colitis [9,10]. 

This observation that endogenous microbes primarily interact with a host indirectly via the production of various compounds has provided further momentum to postbiotic research. Some of these compounds strengthen the epithelial barrier that separates the microbiome from the internal host environment, while other compounds are able to either be absorbed by the epithelium (e.g., in the case of metabolites) or are translocated across the epithelium (e.g., in the case of bigger molecules) into the body where they induce an effect in the internal host environment [11]. Metabolites and secreted products are not unique to a species as many species share the same metabolic pathways. This high degree of functional redundancy within the gut microbiome, together with the great variation in the composition of gut microbiomes of “healthy individuals”, makes it difficult to define a “normal healthy gut microbiome” [12,13]. This highlights the need to shift focus to the preservation of health-associated functionalities instead of attempting to pinpoint specific microbes as biomarkers for a health or disease state [13]. Consequently, microbiome research is now placing increasing attention on the metabolic and secreted products produced by the endogenous microbes as well as the identification and presence of the endogenous microbial species themselves.

The characterisation of microbial species from which postbiotics originate is required by the ISAPP consensus statement as it is essential to confirm that postbiotic products are safe for their intended use [6]. While there is currently no regulatory framework specific to postbiotic preparations for human application, safety assessments are still required to ensure that postbiotic components themselves are not toxic. In order to determine the safety profile of postbiotics, the inactivation method needs to be considered as this is not constrained by the ISAPP postbiotics definition but may impact the type and activity of postbiotic components in a preparation [14]. In order to facilitate the characterisation of postbiotic products, the resulting inanimate postbiotic preparation could be further processed to isolate and analyse particular health-promoting components. The characterisation of the components within postbiotic preparations would enable postbiotic therapies to be developed using standardised concentrations and could thus create a basis for extending and broadening postbiotic research efforts that, to date, have been substantially limited to food-safe microbes targeting the gut microbiome.

## 3. Postbiotics and Human Gut Health

In recent years, deteriorating gut health has been linked to a number of chronic diseases and has been recognised as a key predictor for overall health [15]. There is evidence to suggest that various postbiotic compounds from a broad range of microbes can positively affect several endpoints of gut health [5]. This review focuses on three such compounds and endpoints, namely (i) the impact that specific short-chain fatty acids (SCFAs) have on the health of the gut environment; (ii) the impact that the presence of exopolysaccharides (EPS) can have on strengthening the gut barrier and reducing inflammation within the gut environment; and (iii) the impact that bacteriocins produced by particular bacteria have on promoting antimicrobial activity against gut pathogens. This review also highlights the progress made in certain areas to identify the mechanisms of action inducing these health modulatory effects within the gut microbiome.

The production of the SCFAs butyrate, propionate and acetate by the gut microbiome is important to human health [16,17,18]. SCFAs originate both from the bacterial fermentation of dietary fibres and through cross-feeding in the human colon. SCFAs, among other things, provide protection against intestinal inflammation by activating SCFA-sensing receptors, namely G-protein-coupled receptors (GPCRs), which contribute to intestinal epithelial barrier maintenance and immune regulation [17]. 

Butyrate is the most studied SCFA with significant evidence demonstrating butyrate’s anti-inflammatory and anti-carcinogenic effects [19,20,21]. Butyrate also has a role in overall gut health as it is the primary energy source for the colonic mucosa [17,18,22]. As illustrated in Figure 1, butyrate is produced by a distinct group of gut bacteria via either the CoA-transferase or butyrate kinase pathways [23,24,25]. Butyrate can only be produced by a distinct group of gut microbes, including *Faecalibacterium prausnitzii* and *Eubacterium rectale* [26]. A number of studies have identified that butyrate-producing bacteria are depleted in the gut microbiota of patients with inflammatory bowel diseases (including Crohn’s disease and ulcerative colitis) and colorectal cancer compared to healthy people [21,27]. Furthermore, studies have also demonstrated that butyrate can suppress colonic inflammation and tumour development through the down-regulation of the canonical Wnt-signalling pathway, the inhibition of cancerous colonocyte proliferation through histone deacetylase and the induction of apoptosis in cancer cells [23,28,29,30].

Butyrate-producing bacteria, such as *Faecalibacterium prausnitzii* and *Butyricicoccus pullicaecorum*, are endogenous to the human gut and are associated with a “healthy” human gut microbiome and evidence suggests that the production of butyrate gives *Faecalibacterium prausnitzii* a health promoting effect [8,16,21,25]. This has been illustrated in a study using an in vitro gut model of Crohn’s disease that demonstrated the potential for the supplementation of butyrate-producing bacteria to increase butyrate production, which, in turn, inhibits gut inflammation and enhances intestinal epithelial barrier integrity that is characteristically weak in Crohn’s disease patients [21]. 

Propionate is formed via sugar fermentation by a distinct group of gut bacteria via either the succinate (also termed the randomising pathway), propanediol or acrylate pathway, as illustrated in Figure 1 [23,24,31]. The succinate pathway is the major pathway for propionate production from dietary carbohydrates where hexose and pentose sugars are converted to propionate [23,24,25]. While propionate is less studied than butyrate, there is evidence that the high concentrations of propionate also exert anti-carcinogenic effects with evidence identifying both butyrate and propionate to be the most potent fatty acids to induce differentiation and apoptosis [31]. There is also evidence that propionate lowers lipogenesis and serum cholesterol levels [31]. In addition, a study by Bartolomaeus et al. demonstrated propionate’s role in protection from hypertensive cardiovascular damage using two different mouse models [32].

Acetate is the most abundant SCFA in the gut reaching a molar ratio three times larger than butyrate and propionate [16,26,33]. Acetate is produced through the fermentation of dietary fibres by gut bacteria, including *Ruminococcus* spp., *Prevotella* spp., *Bifidobacterium* spp. and *Akkermansia muciniphila* [33]. Studies have provided evidence that acetate contributes to human health. For example, acetate has been shown to reduce inflammation and insulin sensitivity and improve glucose tolerance by regulating fasting insulin and glucagon levels [33,34,35].

Butyrate- and propionate-producing microbes are currently not considered safe for human use as they are not included on the GRAS or QPS lists. Alternatively, butyrate and propionate-producing bacteria can be used to produce butyrate- and propionate-rich postbiotic preparations that could be administered to humans. However, this is currently an emerging area of research that is not immediately applicable for human therapies given the lack of a regulatory framework specific to postbiotic preparations. Furthermore, further research is needed to better understand uptake of SCFAs in the small intestine and the extent that butyrate- and propionate-rich postbiotic preparations can reach the large intestine.

The use of postbiotic preparations of lactic acid bacteria that are already included on the GRAS and QPS lists is a more attainable approach that is immediately applicable to indirectly increase butyrate and propionate. Lactic acid bacteria are Gram-positive bacteria that share metabolic and physiological characteristics [24]. Most lactic acid bacteria are food-safe and, as such, are good candidates to be used as starter or adjunct cultures in many vegetable-, milk- and cereal-based fermented foods and probiotic supplements [36,37]. The postbiotic preparations of lactic acid bacteria are rich in lactate and acetate that can have a positive down-stream effect on butyrate and propionate concentrations in the gut when converted through cross-feeding by endogenous butyrate and propionate-producers. As illustrated in Figure 1, lactate can be converted to butyrate via the CoA-transferase pathway by microbes, such as *Anaerobutyricum* and *Anaerostipes* species [23,24,26]. Lactate can also be converted to propionate via two pathways, namely the acrylate pathway by microbes, including *Coprococcus catus* and *Megasphaera elsdenii*, or the succinate pathway by microbes, including the *Veillonella* species [23,24,26]. However, where butyrate and propionate producers are not present in the gut, as is the case in some disease states, the increasing concentrations of lactate and acetate will not result in an increase in butyrate or propionate [21,27].

### 3.1. Postbiotics to Strengthen Intestinal Epithelial Barrier Function

The gut barrier is one of the most important barriers between the external environment (including diet, drugs, pathogens and microbiota) and the host. A weakened gut barrier favours the translocation of bacteria and inflammation and has been linked to an increased risk of chronic gut diseases, including Crohn’s disease, ulcerative colitis and irritable bowel syndrome, as well as food-borne infectious diseases, including gastroenteritis caused by pathogens, such as *Salmonella typhimurium* [38,39]. In addition to providing physical protection from potential harmful compounds and pathogen translocation, the intestinal epithelium also acts as a selective barrier for nutrient transport, and for this reason different junction proteins exist to join the epithelial cells to one another and the adjacent tissues (Figure 2) [40]. The various intercellular junctions and junction proteins are collectively referred to as the apical junctional complex [41]. The tight junctions consist of intercellular adhesion protein complexes [41]. The most well studied tight junction proteins are Zonula Occludens-1 (ZO-1) and occludin [41]. Tight junctions are localised closest to the lumen of the intestines and control the diffusion of fluid, electrolytes, macromolecules and prevent the translocation of microbes along the paracellular route [42]. A reduced expression of tight junction proteins ZO-1 and occludin results in a weakening of the tight junctions and correspondingly increases the paracellular permeability, thus allowing uncontrolled translocation of chemicals, antigens or microorganisms across the intestinal epithelium [40]. 

While several studies have demonstrated that compounds produced by a number of lactic acid bacteria have a positive effect on intestinal barrier strength by enhancing the production of the key proteins ZO-1 and occludin, they did not identify the mechanism or the compounds that are responsible for this effect [43,44]. However, using both mice models with induced inflammatory bowel disease and in vitro human Caco-2 cell lines, Zhou et al. provided evidence that EPS from *Lactiplantibacillus plantarum subsp. plantarum* promotes intestinal barrier function by the upregulation of ZO-1 and occludin proteins while repressing the expression of the tight junction protein Claudin-2 and pro-inflammatory cytokines, including IFN-γ, IL-6 and TNF-α [45]. The pro-inflammatory cytokines IFN-γ, IL-6 and TNF- α were shown to be involved in the pathogenesis of inflammatory bowel disease [46,47,48], and the pro-inflammatory cytokine TNF-α also induces the up-regulation of the tight junction protein Claudin-2, which has been shown to contribute to the intestinal epithelial barrier dysfunction in inflammatory bowel disease [45,49]. With respect to the mechanism that resulted in the up-regulation of the ZO-1 and occludin proteins, the study by Zhou et al. highlighted that EPS indirectly activated the transcriptional activator STAT3, which, in turn, bound to the promoter of ZO-1 and occludin [45]. These findings were confirmed using STAT3 knock-down Caco-2 cells where there was no change in intestinal permeability and no change in the expression of ZO-1 and occludin proteins [45]. In summary, this study by Zhou et al. provided evidence that EPS from *Lactiplantibacillus plantarum subsp. plantarum* confers a health benefit by improving the strength of the intestinal barrier function by up-regulating the expression of tight junction proteins and down-regulating the expression of pro-inflammatory cytokines.

The finding by Zhou et al. that the EPS of *Lactiplantibacillus plantarum* subsp. *plantarum* have an intestinal barrier-strengthening quality has been further confirmed to apply to other bacterial-derived EPS in subsequent studies. For example, it has been described that the EPS of *Bacillus subtilis* restored the intestinal barrier integrity of mice models with induced colitis (a type of inflammatory bowel disease) by upregulating the tight junction proteins (occludin, claudin-1 and claudin-2) and downregulating inflammatory cytokines (IL-6 IL-1β) [50]. Similarly, a further study has provided evidence that the EPS of *Streptococcus thermophilus* enhanced expression of tight junction proteins (claudin-1, occludin, and E-canherin) and repressed pro-inflammatory cytokines (interleukin-6 and interferon-γ) [51]. A similar enhanced expression of tight junction proteins and repression of pro-inflammatory cytokines by the EPS of *Streptococcus thermophilus* was also observed using Caco-2 cell lines and led to the conclusion that the EPS of *Streptococcus thermophilus* has the potential to improve intestinal barrier function [51].

### 3.2. Postbiotics to Reduce Inflammation

Chronic gut diseases, such as inflammatory bowel disease and irritable bowel syndrome, are characterised by inflammation and depleted barrier function in the gut and have been associated with oxidative stress resulting from the excessive production of reactive oxygen species (ROS) [52,53,54]. ROS are chemical species formed upon incomplete reduction of oxygen [55]. Free radicals and other ROS are derived from either normal essential metabolic processes in the human body or external sources, such as exposure to X-rays, ozone, cigarette smoking, air pollutants and industrial chemicals [56]. Due to their highly reactive nature, ROS can modify other oxygen species, DNA, proteins or lipids, and excessive amounts of ROS can cause genomic instability [57]. Consequently, the continuous overproduction of ROS compromises gut function, resulting in nutritional malabsorption, increased intestinal permeability and disturbed gut motility [54], and leads to severe tissue injury, resulting in deep ulcers in the ileum, the lower part of the small intestine, as shown in Figure 3 [58]. Serious tissue damage can initiate a chain of events that result in the development or progression of several diseases [59,60,61], including atherosclerosis, arthritis, diabetes, Alzheimer’s disease, neurodegenerative diseases and cardiovascular diseases, as well as inflammatory bowel disease and irritable bowel syndrome [62]. Crohn’s disease patients are characterised by a decreased level of antioxidant activity resulting in severe oxidative stress that is ultimately associated with severe mucosa injury [58,63].

Antioxidants are compounds that scavenge oxygen free radicals or inhibit the oxidation process in a cell [55,63]. Antioxidants thus inhibit oxidative stress by minimising the harmful effects of free radicals on biomolecules and tissues [55]. 

In addition to playing a role in the mechanism that strengthens the intestinal epithelial barrier, EPS from lactic acid bacteria can act as a natural potent antioxidant [60,64,65]. For example, studies have shown that: (i) EPS from lactic acid bacteria have both anti-oxidative and anti-proliferative effects on hepatoma HepG2 cells [66]; (ii) EPS from *Bacillus coagulans* demonstrated significant antioxidant and free radical scavenging activities [59]; (iii) EPS from *Lactiplantibacillus plantarum* subsp. *plantarum* had antioxidant effects that may involve the scavenging of ROS and reduction of lipid peroxidation [61]; and (iv) EPS of lactic acid bacteria, such as *Limosilactobacillus fermentum*, also had antioxidant effects [67]. These studies demonstrate the antioxidant potential of EPS from lactic acid bacteria and indicate that EPS delivered to the gut via postbiotic preparations originating from several lactic acid bacteria could have the potential to induce an antioxidant effect that counteracts ROS. Consequently, EPS has the potential to reduce oxidative stress directly in the gut and thereby positively impact gut and overall health.

### 3.3. Postbiotics with Antimicrobial Activity against Gut Pathogens

Many species of bacteria produce antimicrobial peptides called bacteriocins that are antagonistic against specific microbes [68]. Bacteriocins are produced by the ribosome and exported to the extracellular medium [68], where they are capable of inhibiting pathogens from Gram-negative and/or Gram-positive groups [69]. There are three classes of bacteriocins from lactic acid bacteria. Class I and II bacteriocins are pH and heat stable and thus can still perform their antimicrobial function after being exposed to heat [70], as in the case when preparing postbiotics using heat inactivation. Class III is the only heat labile class of bacteriocins originating from lactic acid bacteria [71], and consequently, Class III bacteriocins only remain active if the postbiotic preparation process does not involve a heat inactivation step. While broad-spectrum bacteriocins have the potential to act against many intestinal pathogens, narrow-spectrum bacteriocins can be used to specifically and selectively inhibit certain pathogens, such as *Listeria monocytogenes*, without affecting beneficial microbes [70]. As bacteriocins have a bactericidal mode of antimicrobial action, usually targeting the cytoplasmic membrane, there is no cross-resistance with antibiotics [70]. 

Research studies have demonstrated that the bacteriocins produced by several bacteria are effective at combatting specific intestinal pathogens. For example: (i) *Latilactobacillus sakei* subsp. *sakei* produces bacteriocins Curvacin A and Sakacin 1, which are both active against *Listeria monocytogenes* [2]; (ii) *Lactiplantibacillus plantarum subsp. plantarum* produces a number of bacteriocins, including Plantaricin L-1, which is active against *Listeria monocytogenes,* as well as Plantaricin MG, which is active against *Listeria monocytogenes* and *Salmonella typhimurium* and also has broad inhibitory activity against Gram-positive bacteria [72,73]; (iii) *Companilactobacillus crustorum* produces bacteriocin BM1157, which displays activity against *Listeria monocytogenes* [2]; (iv) *Lactobacillus gasseri* produces Gassericin A and Gassericin T, which are active against several pathogens, including *Listeria monocytogenes*, *Bacillus cereus* and *Staphylococcus aureus* [74]; and (v) *Ligilactobacillus salivarius* produces a Antilisterial ABP-118, which is active against *Listeria monocytogenes*, as well as *Enterococcus, Bacillus, Listeria, Staphylococcus* and *Salmonella* species [70,75].

These studies demonstrate the potential to incorporate preparations that include bacteriocins originating from lactic acid bacteria within postbiotic-based therapeutics aimed at improving gut health. For example, research has already been conducted to explore the use of bacteriocins as a therapy for the prevention and treatment of adherent-invasive *Escherichia coli* associated inflammatory bowel disease [76]. 

## 4. Postbiotics and Human Health Effects beyond the Gut

Postbiotic research efforts thus far have been mainly directed towards the gut, but the potential application of postbiotics has also been explored in other areas of the human body, including the skin, vagina and the oral cavity.

### 4.1. Postbiotics to Support Skin Health

The skin microbiome has gained increasing research attention as evidence has emerged that alterations within the skin microbiome in particular regions of the body play an important role in a number of skin disorders, including acne vulgaris, eczema, psoriasis and dandruff [77,78]. The skin microbiome is a stable microbial community that exits on the layers of the skin [77,78]. The composition of the skin microbiome differs between body regions [77]. While research into the role of the skin microbiome in the aetiology of skin disorders is still in its infancy, evidence has demonstrated that the skin microbiome plays a key role in maintaining a healthy skin barrier and providing protection against pathogens in addition to altering the immune system [78,79]. Emerging research has demonstrated that altering the skin microbiome can affect certain skin disorders [78]. For example, a mixture of the probiotic strains of *Bifidobacterium lactis*, *Lacticaseibacillus rhamnosus* and *Bifidobacterium longum* was found to be more effective in a randomised controlled clinical trial in treating human patients with psoriasis as a coadjutant treatment together with topical steroids compared to using topical steroids alone [79]. Further, two recent research studies have shown that the postbiotic preparations of *Lactococcus chungangensis* and *Vitreoscilla filiformis* have the potential to positively impact wound-healing in the context of diabetes mellitus and atopic dermatitis, respectively.

Firstly, a study that used type 1 diabetic mice found that wound dressings infused with the postbiotic preparations of *Lactococcus chungangensis* CAU 1447 significantly decreased the size of skin wounds [80]. The postbiotic preparations induced the expression of wound-healing-promoting cytokines, growth factors and chemokines [80]. This study identified a number of specific compounds within the postbiotic preparations of *Lactococcus chungangensis* CAU 1447 that were linked with wound-healing. For example, metabolic analysis identified that palmitic acid and palmitoleic acid are present within the postbiotic preparation of *Lactococcus chungangensis* CAU 1447 [80]. Palmitic acid and palmitoleic acid, which has been shown to increase anti-inflammatory activity and inhibit pro-inflammatory cytokine production, are related to the wound-healing mechanism [81]. Furthermore, this study also identified that stearic acid and linoleic acid, which are involved in vessel formation and tissue regeneration in wound-healing, are present within the postbiotic preparation of *Lactococcus chungangensis* CAU 1447 [80,82].

Secondly, atopic dermatitis is one of several skin disorders that has been associated with an impaired skin barrier and is characterised by disturbed barrier function, skin inflammation and cutaneous dysbiosis [83]. The postbiotic preparations of *Vitreoscilla filiformis* formulated into a moisturiser and applied topically to the skin improved barrier function, promoting the growth of endogenous bacteria and encouraged the restoration of a healthy skin microbiome [84].

### 4.2. Postbiotics to Support Vaginal Health

Lactic acid and D-lactate are acidifying metabolites produced by lactic acid bacteria, dominating a healthy vaginal microbiome. The vaginal acidic conditions, typically ranging between pH 3 and 4.5, created by lactic acid bacteria prevent the colonisation or growth of pathogens that cause common vaginal infections, such as *Gardnerella vaginitis* and *Candida albicans* [85]. Among the probiotic *lactobacilli* strains of vaginal origin, *Lacticaseibacillus rhamnosus* AD3 has been identified as having the most potential for use in an orally administered probiotic therapy as it produces metabolites that encourage the restoration of a healthy vaginal microbiome and that are effective against *Candida* strains [86]. This is consistent with a number of studies that have demonstrated that postbiotic preparations containing the metabolites of *Lacticaseibacillus rhamnosus* (such as lactic and acetic acids) incorporated into a gel and administered intra-vaginally are equally or more effective than antibiotic treatment [85,87,88].

An altered vaginal microbiome characterised by a reduced abundance of *Lactobacillus* species and the presence of the vaginal pathogens *Atopobium vaginae* and *Gardnerella vaginalis* has also been associated with an increased prevalence of preterm labour [85,89,90,91]. Antibiotic therapies that aim to reduce *A. vaginae* and *G. vaginalis* do not appear to be very effective at lowering the risk of preterm birth [90]. However, emerging research into the vaginal microbiome has suggested that the *Lactobacillus* species, particularly *Lactobacillus crispatus*, is protective [90,91]. While the exact mechanism by which *Lactobacillus crispatus* provides protection against preterm labour is currently not fully understood, the pH-lowering effect of the primary metabolic end product of *Lactobacillus crispatus*, namely D-lactate, has been reported to inhibit the growth of vaginal pathogens, such as *Atopobium vaginae* and *Gardnerella vaginalis* [90,92]. In addition, *Lactobacillus crispatus* has immunomodulatory functions that can inhibit the adhesion of bacteria, such as *Gardnerella vaginalis*, to vaginal epithelial cells [93]. This provides evidence to support the potential use of postbiotic therapeutics containing *Lactobacillus crispatus* to tackle preterm birth [90,91]. However, further research is needed to investigate effective methods to apply such preparations rich in D-lactic acid and the associated health risks, given that pregnant women are the target recipients for such biotic therapies.

### 4.3. Postbiotics to Support Oral Health

Alterations in the oral microbiome are associated with a number of oral disorders, including tooth decay, gingivitis and periodontitis [85,94,95]. The oral microbiome is particularly affected by diet and lifestyle. For example, a diet high in sugar lowers the pH of the oral microbiome and, consequently, creates the conditions for various acid-tolerant microbes to thrive, including *Streptococcus mutans*, which has been associated with the aetiology of tooth decay [96]. Once established, *S. mutans* produces acids that demineralise teeth making them more susceptible to infections and cavities. Moreover, the acids further contributes to the low pH conditions in which *S. mutan* thrives [96]. This evidence linking certain microbes to certain oral disorders suggests that biotic therapies (including probiotic, prebiotic and postbiotics) which alter the oral microbiome have the potential to treat many oral disorders. Research is ongoing to investigate whether biotics may be used to modulate a disordered oral microbiome. For example, a study demonstrated that probiotic *Lactococcus lactis* was successful in preventing and disrupting the formation of pathogen-spiked oral biofilms as well as decreasing the abundance of pathogens and thus promoting a healthier oral microbiome [94]. A further study demonstrated that Nisin, a bacteriocin produced by *Lactococcus lactis*, exhibited anti-biofilm and antimicrobial effects against a number of oral pathogens, including *Porphyromonas gingivalis, Prevotella intermedia, Aggregatibacter actinomycetemcomitans, Fusobacterium nucleatum* and *Treponema denticola* [95]. As such, these findings suggest that the postbiotic preparations of *Lactococcus lactis* that include the bacteriocin Nisin have the potential to be used in a postbiotic therapy for human application to support a healthy oral microbiome.

## 5. Postbiotics and Their Potential Use in Commercial Products

One of the main potential advantages of using postbiotic preparations in human therapeutics is that the microbial biomass is inanimate. Hence, postbiotics should not be constrained to the same health and safety measures that have been designed for products that include live microbes. Consequently, this opens up the opportunity to incorporate postbiotics from microbes that are not on QPS or GRAS lists into therapeutics for human application. The studies described in this review illustrates that while research into the health benefits of postbiotics is still in its infancy, there is considerable potential for using postbiotics in commercial products in order to provide health benefits to a consumer.

Currently, the main area where probiotics are being used in gut-health-promoting commercial products are fermented foods, even though the vast majority of consumers are unaware of the presence of microbial components. Traditional fermentation uses wild microbes to achieve leavening and improve both the preservation and flavour of food products through secretion of a range of metabolites during the fermentation process. As the microbes that are involved in the traditional fermentation of food products have a history of safe use there is a high level of confidence for using these microbes in human therapeutics. While the identity, presence and potential health benefits of the vast majority of these metabolic and secreted products are unknown, it has become increasingly recognised that consuming fermented foods, such as sauerkraut and kimchi, may offer consumers a potential health benefit [97]. Despite this, products based on these traditional fermentation processes have become less common now that food products can be easily stored using refrigeration and leavened using both chemical rising agents and commercial yeast preparations. 

In recent years, however, the demand for traditional fermented products, such as sourdough bread, has increased. While the increase in the popularity of sourdough bread is mainly due to its enhanced flavour, there are also reports of health benefits associated with sourdough bread [97]. Sourdough bread is baked at a high temperature, inactivating live microbes that ferment the bread dough and thus any health benefits from consuming sourdough bread come from the cell fragments and/or the heat stable metabolic and secreted compounds that are produced during the fermentation of the bread dough. While these cell fragments and/or the heat stable metabolic and secreted compounds have previously often been referred to as “postbiotics” and contain many of the characteristics of postbiotics under the ISAPP definition, the ISAPP definition specifically excludes such products originating from wild microbes that are neither characterised nor defined [6]. Consequently, either the wild microbes have to be characterised or the fermentation needs to be carried out using a defined set of characterised microbes in order for traditional fermentation to be considered as postbiotic preparations [6].

Examples of commercial products that aim to improve gut health include Colibiogen^®^ oral and Hylak^®^ Forte [98]. Colibiogen^®^ oral (Laves-Arzneimittel GmbH, Ronnenberg, Germany) is a commercial product that is based on a cell-free filtrate derived from the cultures of *Escherichia coli* (strain Laves 1931), containing several metabolic products, including amino acids, peptides, polysaccharides and fatty acids [99]. After a clinical trial was completed in 2016 with a positive outcome, Colibiogen^®^ oral has been marketed for irritable bowel syndrome patients with a dysfunction of the intestinal mucous membrane barrier [99]. Hylak^®^ Forte (Ratiopharm/Merckle GmbH, Vienna, Austria) contains the metabolites of four bacterial strains, namely *Lactobacillus acidophilus* (DSM 4149), *Lactobacillus helveticus* (DSM 4183), *Escherichia coli* (DSM 4087) and *Enterococcus faecalis* (DSM 4086), and has been designed to inhibit the growth of pathogenic bacteria by reducing the intestinal pH [100]. A study in 2014 provided evidence that Hylak^®^ forte was highly effective in the treatment of intestinal dysbacteriosis of patients with chronic gastritis and supported the prolonged use of Hylak^®^ forte to prevent relapses of chronic gastritis [101]. While products such as Colibiogen^®^ oral and Hylak^®^ forte could have been considered to contain postbiotics under some previous definitions prior to 2021, such products would not fall within the definition of a postbiotic preparation as set out in the recent 2021 ISAPP consensus statement, which now requires cellular biomass to be present within a postbiotic preparation. Instead, despite evidence of health-promoting effects, products that include cell-free filtrates and isolated compounds should now be referred to using their chemical name without any reference to postbiotics. For example, where a supplement includes lactic acid isolated from a postbiotic preparation, the ISAPP consensus statement indicates that this should be labelled as “contains lactic acid” rather than “contains isolated postbiotics”.

Beyond this, several other isolated metabolic and secreted products from food-safe microbes are currently purposely added to food products to improve their shelf-life or texture. For example, EPS produced from lactic acid bacteria is already being used to enhance the texture of various commercial food products, such as gluten-free bread and yogurt [102,103,104,105]. In addition, Nisin is widely used as a bio-preservative and is thereby the most prominent postbiotic-derived commercially applied compound. Nisin is a bacteriocin produced by a group of Gram-positive bacteria that belong to *Lactococcus* and *Streptococcus* species [106]. Consequently, as several of the studies referred to above highlight the potential of both EPS and bacteriocins to induce health effects within the gut microbiome, further research is needed. Specifically, investigating whether and to what extent the EPS and bacteriocins (specifically Nisin) of specific microbes could both provide health effects to a consumer if administered as part of a postbiotic preparation, in addition to continuing to effectively perform their original function to improve the shelf-life or texture of food products.

## 6. Knowledge Gaps and Potential Future Applications

From the many studies referred to in this review, it is clear that researchers have begun to identify specific components within postbiotic preparations that play a key role in producing health-modulatory effects. However, further research needs to be conducted to address knowledge gaps and to broaden the understanding how these health modulatory effects are induced. 

Firstly, there is a need to create a dedicated reporting platform that enables researchers to share details of the components of postbiotic preparations, the applied inactivation method and the microbes that produce them. This will enable the consolidation of research findings and facilitate the commercialisation of products. The identification of novel postbiotics will require a shift in research focus beyond the presence of specific microbes and towards the postbiotic components of these microbes that are associated with specific health promoting qualities in each of the gut, skin, vaginal and oral microbiomes. This would provide the foundation to be able to define a “healthy microbiome environment” by the presence of a particular set of key postbiotic component biomarkers. Furthermore, to understand how these health effects are produced, additional research efforts are also needed to investigate the mechanisms and mode of action by which specific postbiotic components deliver direct improvements in gut health and/or lead to other indirect health benefits within a human host.

Secondly, in contrast to probiotics, postbiotics do not proliferate or grow; thus, they are not able to provide a sustained effect once their supplementation stops. Additional research efforts will therefore be needed to identify a range of minimal concentrations of the key postbiotic biomarkers required to induce a positive health effect. The quantification of postbiotic biomarkers will also help to determine whether the levels detected within a particular host microbiome environment are deficient and/or indicative of a disease state and whether a host may thus benefit from postbiotic supplementation.

Thirdly, where supplementation of key postbiotic compounds is considered beneficial, further research efforts will also be required to determine the extent to which supplemented postbiotics delivered through different mechanisms actually reach the target microbiome and have the desired impact. While this may be more straightforward with regards to the skin, vaginal and oral microbiomes, which are relatively easily accessible, this will be considerably more complex when considering the gut microbiome. In this case, research using in vitro gut models will likely play a crucial role in this effort.

In addition, given that the 2021 ISAPP postbiotic definition requires all microbes and microbial products in postbiotic preparations to be inanimate, further work needs to be conducted to both expand research to include non-food-safe microbes. Furthermore, the focus of postbiotic research can now be extended to a much larger population of non-food-safe microbes and beyond the narrower range of food safe microbes that were formerly the focus of probiotic research, including endogenous microbes. This includes considering the health promoting effects that could be obtained using postbiotic preparations from the non-food-safe endogenous microbes within a microbiome but could equally include postbiotic preparations using microbes exogenous to a particular microbiome that have the potential to producing particular health promoting effects. For example, there is the opportunity to investigate whether particular postbiotic components associated with specific health-promoting qualities in one microbiome environment have the potential to induce similar desirable effects in other distinct microbiome environments. For example, each of the gut, skin, vaginal and oral microbiomes have a common feature in that they all exist in close proximity to an epithelial barrier and that the epithelial barrier plays a key role in the maintenance of host health by protecting and shielding the internal host against the external environment. As a result, when the epithelial barrier function is disturbed a number of diseases or disorders that are associated with epithelial barrier weakness can develop [107]. Although research into the skin, vaginal and oral microbiomes is not as advanced as the gut microbiome, insights gained from epithelial barrier strengthening postbiotic therapies as result from gut microbiome research could be transferrable for use across the other microbiomes. 

Finally, as postbiotic preparations cannot contain live microbes, the regulatory frameworks for probiotics set up by the FDA and EFSA based on the GRAS and QPS lists, respectively, do not apply to postbiotics. Consequently, this appears to create a regulatory vacuum that gives a considerable degree of freedom in product commercialisation and development involving postbiotic preparations beyond the minimum requirement to ensure that postbiotic components themselves are not toxic. As a result, research needs to be conducted to consider an appropriate set of regulatory and safety parameters that should be applied to postbiotic preparations until the FDA and EFSA are able to develop a regulatory framework specifically for postbiotics.

## 7. Conclusions

While still in its infancy, postbiotic research has built upon and expanded the understanding gained from probiotic research efforts to further investigate and explain how alterations in the human microbiome environments can impact human health and disease and is providing a research base upon which to develop postbiotic therapeutics. However, the recent publication of the 2021 ISAPP consensus statement, including the definition and scope of postbiotics, reflects the need to agree to a common understanding of the distinctive characteristics of postbiotics among the research community and provide a solid basis for future research efforts in order to address knowledge gaps.

## Figures and Tables

**Figure 1 biomolecules-12-01640-f001:**
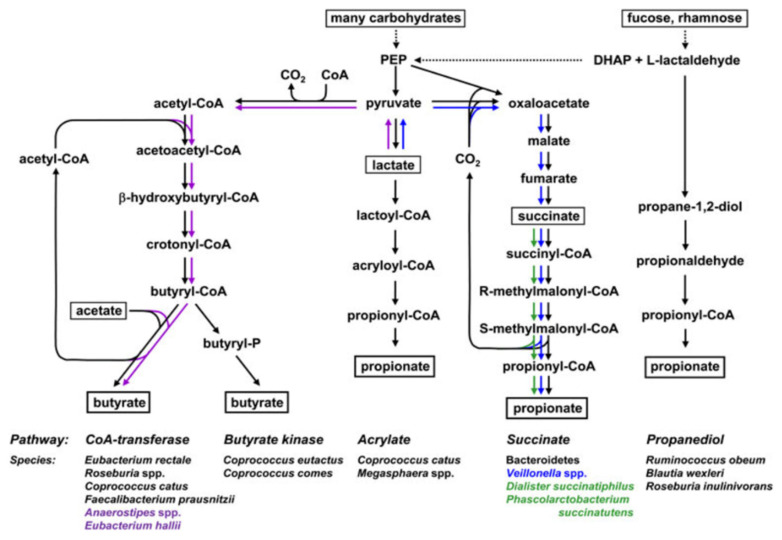
Metabolic routes for butyrate and propionate formation by representative bacterial genera and species from the human colon. Species shown in purple can utilise lactate to form butyrate; species shown in blue and green can, respectively, utilise lactate and succinate to produce proprionate. DHAP, dihydroxyacetonephosphate; PEP, phosphoenolpyruvate. Figure reprinted from Flint, Duncan et al., 2015 [23].

**Figure 2 biomolecules-12-01640-f002:**
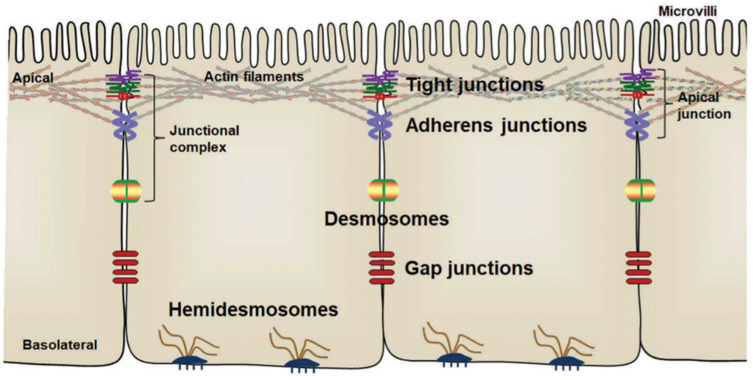
Arrangement of intestinal epithelial cells and intercellular junctions between epithelial cells. The apical junctions are composed of tight junctions and adherens junctions. Figure reprinted from Zhu, Sun and Du. 2018 [40].

**Figure 3 biomolecules-12-01640-f003:**
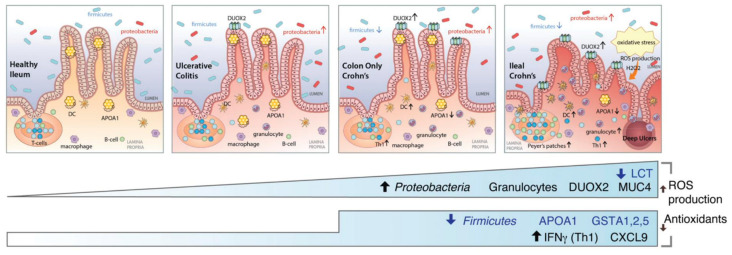
Host gene expression and microbial shifts across the spectrum of ileal IBD. Figure reprinted from Haberman, Tickle et al., 2014 [58].

## Data Availability

Not applicable.

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
