# Peer review of "Postbiotics and Their Health Modulatory Biomolecules"

_biomolecules, 2022, doi:10.3390/biom12111640_

Round 1

Reviewer 1 Report

The subject of the manuscript is suitable for the journal and the Special Issue in particular. The review is structured well and extensive. It covers a relevant topic and the described Knowledge gaps and potential future applications are very useful for scientist working the filed. The references are relevant (a high number of citations from the last 5 years),

I recommend publication in present form.

I am available for further review, if needed.

Author Response

Thank you for your comments and feedback. 

Reviewer 2 Report

This review starts from the definition of Postbiotics reported in the position paper of ISAPP published in 2021, and sets out the current research on the effects of postbiotics on human health. The use of commercial products containing probiotics is also reported.

This manuscript is well-written, easy to follow and includes an updated and critical review of the scientific literature on this very interesting hot topic.

I would like to suggest some minor revisions:

- The Authors should avoid citing references in the Abstract

- A thorough revision of the cited references is mandatory, as several mistake are present. In addition, when possible, the Authors should cite original articles reporting study results instead that review papers.

The References section need to be carefully revised.

- § 2 Page. 2: The Authors stated that “Components of inactivated cells on their own have been termed ‘paraprobiotics’ [7].” This definition is not coherent with the definition reported in the cited paper, which is as follows: “PARAPROBIOTICS: non-viable microbial cells (either intact or broken) or crude cell extracts which when administered (either orally or topically) in adequate amounts, confer a benefit on the human or animal consumer)”. Could the Authors check this point?

- § 3.2. page 5: References 38-43 are not reported in the text, while refencence 3,4,5,7 that do not seem to be pertinent to the text. The Authors should correct the reference numbering. The following sentence should be rephrased : “The tight junctions consisting of intercellular adhesion protein complexes, including Zonula Occludens-1 (ZO-1) and occludin being the more well studied and easiest junction to study are best studied in relation to gut barrier strength”.

- § 3.3. Page 6: “or external sources such as exposure to X-rays, ozone, cigarette smoking, air pollutants, and industrial chemicals [63].  Ref 63 is OK but the reference numbering is wrong.

- § 3.3. Page 7: Could the Authors check the references 58-61 and 67? They do not seem to be pertinent to the text. Ref 67 should be also checked.

- § 3.4 page 7: References 69 and 70 are the same paper.

- § 5 page 9-10: In my opinion, this paragraph could be shortened as several sentences are not completely functional to the topic of the review. For instance, the Authors could consider to modify the central paragraph of page 10 (Currently….). Ref 99 do not deals with postbiotics and does not properly support the text.

Author Response

Thank you for your comments and feedback. I have addressed the revisions, namely avoiding citing references in the abstract, revising cited references. I provide a point by point response to each of the comments below.

Minor revisions suggested:

  • The Authors should avoid citing references in the Abstract 
    • I have removed citing references in the abstract.
  • A thorough revision of the cited references is mandatory, as several mistake are present. In addition, when possible, the Authors should cite original articles reporting study results instead that review papers.
    • I have updated references and double checked all the references in the manuscript.

The References section need to be carefully revised.

  • § 2 Page. 2: The Authors stated that “Components of inactivated cells on their own have been termed ‘paraprobiotics’ [7].” This definition is not coherent with the definition reported in the cited paper, which is as follows: “PARAPROBIOTICS: non-viable microbial cells (either intact or broken) or crude cell extracts which when administered (either orally or topically) in adequate amounts, confer a benefit on the human or animal consumer)”. Could the Authors check this point?
    • I see the misunderstanding and removed I have deleted, “which had previously been referred to in some studies as ‘paraprobiotics’" from the text. 
  • § 3.2. page 5: References 38-43 are not reported in the text, while refencence 3,4,5,7 that do not seem to be pertinent to the text. The Authors should correct the reference numbering. The following sentence should be rephrased : “The tight junctions consisting of intercellular adhesion protein complexes, including Zonula Occludens-1 (ZO-1) and occludin being the more well studied and easiest junction to study are best studied in relation to gut barrier strength”.
    • I have updated references so that 38-42 is reported in the text
    • I have rephrased that sentence to “Tight junctions consist of intercellular adhesion protein complexes. The most well studied tight junction proteins are Zonula Occludens-1 (ZO-1) and occludin”
    • It now reads “Tight junctions consist of intercellular adhesion protein complexes. The most well studied tight junction proteins are Zonula Occludens-1 (ZO-1) and occludin.”
  • § 3.3. Page 6: “or external sources such as exposure to X-rays, ozone, cigarette smoking, air pollutants, and industrial chemicals [63].  Ref 63 is OK but the reference numbering is wrong.
    • I have updated the reference numbering. 
  • § 3.3. Page 7: Could the Authors check the references 58-61 and 67? They do not seem to be pertinent to the text. Ref 67 should be also checked.
    • I have checked and updated the references.
  • § 3.4 page 7: References 69 and 70 are the same paper.
    • I have amended this error so that there is only one reference to the same paper.
  • § 5 page 9-10: In my opinion, this paragraph could be shortened as several sentences are not completely functional to the topic of the review. For instance, the Authors could consider to modify the central paragraph of page 10 (Currently….). Ref 99 do not deals with postbiotics and does not properly support the text.
    • I have shorted the paragraph on pages 9-10 and removed ref 99.